# EXPLORING MINIMUM BAYES RISK DECODING FOR TEXT-TO-SQL ENSEMBLE

## ABSTRACT

The task of translating natural language into SQL (NL2SQL or text-to-SQL) enables users to query relational databases without requiring SQL expertise. Although recent large language model (LLM) approaches have advanced the field, achieving robust performance continues to depend on ensemble methods. Existing heuristic-based ensembles such as Minimum Bayes Risk (MBR) and Model-Based MBR (MBMBR) either ignore model-predicted probabilities or allow low-probability candidates to dominate the selection process, and they suffer from prompt sensitivity when estimating candidate likelihoods. We propose a novel heuristic-based ensemble method that directly incorporates each candidate's own probability into its heuristic score while mitigating prompt sensitivity through marginal probability estimation across diverse prompts. This formulation both improves traditional MBR and stabilizes probability estimation, enabling more accurate and higher-performing candidate selection without the computational overhead of supervised or prompt-based ensembles. Extensive experiments on the SPIDER and BIRD benchmarks demonstrate that our approach consistently outperforms state-of-the-art heuristic methods, achieving higher execution accuracy across fine-tuned and pretrained LLMs. Ablation studies confirm that both the probabilistic scoring function and the marginal probability estimation independently contribute to performance gains, with the full method delivering the strongest results. Our findings establish a new state of the art for heuristic-based ensembles in NL2SQL and highlight the broader potential of probability-aware ensemble strategies for natural language generation tasks.

## 1 INTRODUCTION

The task of translating Natural Language to SQL (NL2SQL or text-to-SQL) enables users to express queries in natural language and automatically obtain the corresponding SQL code (Yu et al., 2018; Zhong et al., 2017). This task typically involves generating SQL queries conditioned on a database schema and natural language question, often leveraging metadata or example values from the database (Li et al., 2024b; Gao et al., 2024). As the volume of data continues to grow exponentially (Huberman & Adamic, 1999), efficient data access has become increasingly critical. Since most organizational data is stored in relational databases and accessed via SQL, users without technical expertise face significant barriers in retrieving information. Thus, NL2SQL systems play a vital role in reducing the cost and time of data access by empowering non-technical users and accelerating query generation for technical users. Despite notable progress, however, current systems still lag behind human experts (Shkapenyuk et al., 2025; Pourreza et al., 2024; Gao et al., 2024), highlighting the need for more robust and generalizable approaches.

Many previous approaches generally produced a single SQL, either by fine-tuning models on task-specific datasets (Li et al., 2024b) or by prompting pretrained LLMs (Pourreza & Rafiei, 2023; Gao et al., 2023), using decoding approaches such as sampling, beam search, or greedy decoding. However, restricting the system to a single candidate often results in suboptimal outputs, as decoding does not guarantee the best solution. Recent work has shown that generating multiple candidate queries and then selecting the most promising one significantly improves accuracy (Pourreza et al., 2024; Gao et al., 2024), a strategy that has proven effective not only in NL2SQL but also in broader code generation tasks (Shi et al., 2022).

Existing ensemble methods for candidate selection can be grouped into three main categories: (1) Supervised ensemble models (Jiang et al., 2023; Gao et al., 2024; Gorti et al., 2024; Pourreza et al., 2024), which require expensive training and tend to generalize poorly across domains and candidate generation methods. (2) Prompt-based ensemble methods (Du et al., 2023; Talaei et al., 2024; Sheng et al., 2025), which query an LLM to select the best candidate. While effective, they rely on large LLMs at inference time and incur high computational costs due to multi-step comparisons and few-shot prompting. (3) Heuristic based methods (Shi et al., 2022; Jinnai et al., 2023; Li et al., 2024a), which avoid training and inference costs but typically underperform in terms of accuracy.

In this work, we focus on heuristic-based ensembles for their efficiency. Existing methods, such as voting and Minimum Bayes Risk (MBR), typically ignore the model's predicted probabilities, relying instead on execution results(Li et al., 2024a; Sheng & Xu, 2025) or surface-form similarity (Shi et al., 2022). Model-Based MBR (MBMBR) partially addresses this by incorporating candidate probabilities into the selection heuristics (Jinnai et al., 2023), but it still has two limitations: the heuristic score of a candidate is largely determined by the probabilities of other candidates, which can overshadow the candidate's own probability, and the method is highly sensitive to prompt variations, leading to unstable selection.

We propose a novel heuristic-based ensemble method that addresses both limitations. Our approach incorporates each candidate's probability directly into its heuristic score while also weighting pairwise similarity by the probabilities of both candidates. To mitigate prompt sensitivity, candidate probabilities are estimated using multiple prompts, marginalizing out noise from prompt context. Compared to supervised or prompt-based methods, this approach avoids expensive training or multi-step inference, while improving candidate selection reliability and overall accuracy.

Through extensive experiments, we demonstrate that our method consistently outperforms previous heuristic-based ensembles across multiple NL2SQL benchmarks. Our ablation studies confirm that the probabilistic scoring function and the marginal probability estimation each contribute independently to performance gains, while their combination yields even larger improvements by reinforcing each other's effect. We further analyze performance across question difficulty levels and varying numbers of candidates, showing that our method consistently outperforms baselines. Together, these results establish a new state of the art for heuristic-based candidate selection in NL2SQL and highlight the broader potential of principled ensemble strategies in natural language generation tasks.

## 2 RELATED WORK

**Ensemble Methods.** Early ensemble methods were introduced to enhance the performance of individual models by aggregating the outputs of multiple models. Classical techniques include bagging, boosting, and stacking (Breiman, 1996a;b; Wolpert, 1992). In bagging, several homogeneous models are trained in parallel on bootstrapped subsets of training set, and their outputs are aggregated by voting or averaging (Breiman, 1996a). Boosting sequentially trains models, giving higher weight to previously misclassified samples to correct past errors (Breiman, 1996b). Stacking combines heterogeneous models by training a meta-model (Wolpert, 1992). All these classical methods require independently trained components to introduce diversity into the ensemble.

Recent work has explored more sophisticated techniques for leveraging multiple models. One line of research focuses on model merging, where the parameters of homogeneous models are combined into a single model before inference (Yang et al., 2024; Cohere et al., 2025). Although this approach avoids ensemble predictions at inference time, it requires extensive training and lacks interpretability and flexibility. Furthermore, merging is limited to homogeneous models. To address heterogeneity, DEEPEN maps the latent spaces of different large language models (LLMs) into a relative space and averages them (Huang et al., 2024). However, this also sacrifices interpretability and modularity.

Some other approaches operate on the intermediate generation steps to retain interpretability. For instance, EBBS performs token-level ensembling by averaging token probabilities and trimming the distribution tails (Wen et al., 2025). Also, Li et al. (2023) introduce the Step-Aware Verifier that evaluates intermediate reasoning steps for each candidate reasoning path. However, EBBS becomes ineffective when LLMs are overly confident as in code generation, where the next token is more deterministic. The Step-Aware Verifier, meanwhile, is confined to reasoning tasks with discrete, interpretable steps, such as arithmetic.

In another related line of work, the ensemble is performed on the final output of each model using some heuristics. For instance, GenQREnsemble generates multiple candidates and concatenates them into a new reformulation of the original query (Dhole & Agichtein, 2024). Another example is MBR-EXEC (Shi et al., 2022), Shi et al. (2022) propose Minimum Bayes Risk (MBR)-based ensemble selection method that utilize execution feedback, demonstrating strong performance in code and SQL generation when such feedback is available. Inspired by this, we adopt execution feedback and further enhance the MBR scoring function by incorporating candidate probabilities, resulting in improved ensemble performance.

Some prior works also explore modifications of the MBR framework. For example, Jinnai et al. (2024) introduces MBR as a proximity regularizer for Best-of-N decoding to mitigate reward hacking. Zhang et al. (2022) propose a regularized MBR re-ranking strategy using multiple regularizers, but report limited and inconsistent gains. More closely related to our work, Jinnai et al. (2023) present a model-based MBR (MBMBR) formulation that integrates candidate probabilities. We build upon this by refining the MBMBR approach further, leading to improved ensemble effectiveness in the text-to-SQL setting.

**Ensemble in Text-to-SQL.** Although some of the text-to-SQL methods do not use any ensemble in their work (Gao et al., 2023), having some form of ensemble is very common among all text-to-SQL approaches. To the best of our knowledge, all the ensemble methods used in the previous text-to-SQL work are applied to the final outputs rather than the intermediate steps. While some approaches only rely on simple techniques such as executability check (Li et al., 2024b) or self-consistency (Shkapenyuk et al., 2025; Sheng & Xu, 2025; Xie et al., 2025; Li et al., 2024a), there is a large body of work leveraging more advanced ensemble approaches to improve the end-to-end performance of text-to-SQL conversion.

Among these advanced methods, some approaches fine-tune a model to generate the final output based on the outputs from all components. For example, LLM-Blender first trains a model to compare pairs of candidates and then uses another fine-tuned model to fuse a subset of selected candidates into the final answer (Jiang et al., 2023). Similarly, many text-to-SQL approaches fine-tune models to select the best candidate (Gorti et al., 2024; Pourreza et al., 2024; Dönder et al., 2025; Gao et al., 2024) or to predict certain properties of the most promising candidate (Zeng et al., 2023). However, training such classifiers is computationally expensive and hinders the ability to generalize to new domains. In contrast, our approach is fully unsupervised and requires no training.

Another set of ensemble approaches use prompting techniques with pre-trained LLMs to choose the best candidate. In text-to-SQL, several prompting strategies have been introduced, including binary classification (Cao et al., 2024), merge-and-revise (Sheng et al., 2025), multiple-choice prompts (Lee et al., 2024), and unit test generation (Talaei et al., 2024). Although prompting-based approaches are generally less expensive than fine-tuning, they require large context windows for in-context demonstrations and substantial hardware resources to deploy powerful LLMs, especially since smaller models often underperform on these tasks.

## 3 METHOD

### 3.1 CLASSIC MBR AND MBMBR SETUPS

Minimum Bayes Risk (MBR) and its variant, Model-Based Minimum Bayes Risk (MBMBR), are common approaches for selecting the best hypothesis from a set of candidates by maximizing expected utility, which equivalently corresponds to minimizing the associated risk. Given a candidate set $\mathcal{H}$ and a utility function $u(h, y)$ that measures the similarity between two hypotheses $h$ and $y$, the goal is to select the hypothesis with the highest expected utility. Concretely, for each candidate, MBR and MBMBR estimate its expected similarity to the remaining candidates under the distribution $P(y)$, and the selected hypothesis is the one with the greatest expected similarity. The scoring rule for both MBR and MBMBR follows this general structure:

$$h^* = \arg\max_{h \in \mathcal{H}} \mathbb{E}_{y \sim \mathcal{H}}[u(h, y)] = \arg\max_{h \in \mathcal{H}} \sum_{y \in \mathcal{H}} u(h, y) \cdot P(y) \tag{1}$$

In standard MBR, the probability $P(y)$ is assumed to be uniform over the candidates, since they are typically generated via Monte Carlo sampling. This leads to a simplified scoring function:

$$h^* = \arg \max_{h \in \mathcal{H}} \sum_{y \in \mathcal{H}} u(h, y) \tag{2}$$

MBMBR improves upon this by using model-predicted probabilities $P(y)$ instead of assuming a uniform distribution. However, both MBR and MBMBR share a critical limitation: the probability of a candidate $h$ does not directly contribute to its own score. This is because the scoring function conditions on $h$ when computing the expectation of the utility, and thus $P(h)$ does not appear in the calculation except in the trivial case when $h$ is compared with itself through $u(h, h)$. Instead, the probability of other candidates influence the score of $h$ based on how similar they are to $h$. This creates the potential for reward hacking, where a low-probability candidate that happens to maximize expected utility is selected, even if its own likelihood is implausibly low.

For instance, consider three candidates $\mathcal{H} = A, B, C$ with utility scores $u(A, B) = 0.9$ and $u(A, C) = u(B, C) = 1$, and where $u(X, X) = 1$ for any $X \in A, B, C$. Let the probabilities be $P(A) = P(B) = 0.5$ and $P(C) = 0$. Applying the MBMBR scoring formula yields Score$(A) = 0.95$, Score$(B) = 0.95$, and Score$(C) = 1.0$. In this setting, $C$, despite having zero probability, receives the highest score because it has perfect utility similarity with the high probability candidates $A$ and $B$, resulting in its incorrect selection as the best hypothesis.

### 3.2 OUR PROPOSED METHOD

In this section, we introduce two complementary techniques that address key limitations in candidate selection and probability estimation when using language models. First, we present our **Scoring Method** (Risk Minimization Function), a refined decision rule that incorporates each candidate's own model-assigned probability into its utility score, thereby addressing the risk of selecting implausible outputs. Second, we describe our **Marginal Probability Calculation** approach, which mitigates prompt sensitivity bias by marginalizing over diverse few-shot demonstrations and applying structured length normalization. Together, these methods improve both the robustness and the reliability of model-based candidate evaluation.

#### 3.2.1 OUR SCORING METHOD (RISK MINIMIZATION FUNCTION)

To mitigate the issue mentioned in subsection 3.1, we propose a probabilistic scoring function that explicitly accounts for the contribution of each candidate to the total expected utility. Instead of assuming that a specific candidate is given, we compute the difference in total expected utility with and without a candidate in the set. This results in the following scoring rule:

$$h^* = \arg \max_{h \in \mathcal{H}} \left( \mathbb{E}_{x, y \sim \mathcal{H}}[u(x, y)] - \mathbb{E}_{x, y \sim \mathcal{H} \setminus \{h\}}[u(x, y)] \right) \tag{3}$$

In Eq. (3), the first term $\mathbb{E}_{x, y \sim \mathcal{H}}[u(x, y)]$ represents the total expected utility across all candidates in the pool, which remains constant regardless of the specific candidate under consideration. The second term, $\mathbb{E}_{x, y \sim \mathcal{H} \setminus h}[u(x, y)]$, corresponds to the total expected utility when candidate $h$ is removed from the pool. Importantly, neither of these terms is conditioned on any specific candidate, ensuring that the probability of every candidate contributes to the computation of the expectation. In other words, the scoring rule evaluates which candidate's absence leads to the greatest reduction in the overall expected utility. You can find the simplified form of this scoring rule below, and its proof is provided in Appendix B.

$$h^* = \arg \max_{h \in \mathcal{H}} \left( P(h) \sum_{y \in \mathcal{H}} u(h, y) \cdot P(y) - \frac{1}{2} \cdot P(h)^2 \right) \tag{4}$$

In the simplified scoring rule in Eq (4), the influence of $P(h)$ on a candidate's own score becomes explicit. The first term, $P(h) \sum_{y \in \mathcal{H}} u(h, y) \cdot P(y)$, is equivalent to the MBMBR formulation in Eq. (1), scaled by the probability of the candidate itself. The second term, $-\frac{1}{2} P(h)^2$, while typically small in practice, serves to counterbalance the potentially disproportionate emphasis on $P(h)$

introduced by the first term. Moreover, the classical MBR scoring rule (Eq. (2)) can be recovered as a special case of our formulation by assuming a uniform distribution over candidates. Thus, our proposed method generalizes the MBR framework while addressing its limitations.

### 3.2.2 UTILITY FUNCTION MOTIVATION AND HYPER-PARAMETER SELECTION

To instantiate the utility function $u(h, y)$, we define it based on the execution results of the hypotheses $h$ and $y$. Let $H$ and $Y$ denote the execution results of $h$ and $y$, respectively. Let $\phi(H)$ be a binary indicator function that returns 1 if $H$ is empty or results in an execution error, and 0 otherwise. Let $\eta(H)$ be the fraction of rows in $H$ that contain NULL values. The utility function is then defined as:

$$u(h, y) = \begin{cases} e^{-2} & \text{if } \phi(H) = 1 \\ e^{-1} & \text{if } \eta(H) > \epsilon \\ e^{\lambda \cdot \text{Jacc}(H,Y)} & \text{otherwise} \end{cases} \quad (5)$$

Here, $\text{Jacc}(H, Y)$ denotes the Jaccard similarity between the execution results $H$ and $Y$. The threshold $\epsilon$ specifies the allowable proportion of NULL values . The hyperparameter $\lambda$ controls the influence of the utility score relative to candidate probabilities in the probabilistic scoring function.

**Motivation for the Piecewise Exponential Design.** The structure of the utility function reflects the semantics of SQL execution results. (1) Empty outputs and execution errors represent complete failure and should receive uniformly low utility, (2) partially valid outputs (high NULL ratios) represent degraded results, and (3) valid outputs warrant a continuous similarity-based score. A piecewise function therefore matches the discrete nature of execution correctness. We use exponentials because they integrate cleanly with log-probabilities used during decoding and naturally create smooth yet well-separated utility scales.

**Rationale Behind $e^{-2}$ and $e^{-1}$.** These constants are not tuned hyperparameters but fixed penalties intended only to enforce the ordering: (1) errors $<$ (2) partial results $<$ (3) valid results. We verified that changing these constants yields nearly identical performance as long as the ordering is preserved. This demonstrates that the method is not sensitive to the exact numerical values.

**Threshold $\epsilon$.** The threshold $\epsilon$ is not a tunable hyperparameter in our method. Instead, it is derived empirically from the characteristics of the BIRD training set. Specifically, the execution results of many ground-truth queries in the dataset contain some NULL values. We therefore inspected the training data and observed that having fewer than approximately %20 NULL values is common for correct, executable queries, whereas higher proportions usually correlate strongly with invalid SQL predictions. Therefore, we fixed $\epsilon = 0.2$ for all models, datasets, and experiments. Since this choice reflects a stable property of the underlying data rather than model-dependent tuning, $\epsilon$ is identical across all experiments and was never optimized on validation or test sets.

**Scaling parameter $\lambda$.** $\lambda$ is the only hyperparameter that is tuned in our method. Its role is twofold: (1) Matching the scales of probabilities and Jaccard similarity. The Jaccard similarity lies in the range [0,1], while candidate probabilities vary significantly across models and datasets. Larger models tend to be more confident and assign higher peak probabilities, and the scale changes further when using marginal probability, where candidate scores are aggregated through summation rather than maximum-likelihood. Because these probability scales differ, $\lambda$ is needed to place the Jaccard similarity and probability terms on comparable numerical ranges so that neither term dominates the final ensemble score. (2) Controlling the tradeoff between probability and execution similarity. $\lambda$ also regulates how the scoring function prioritizes model confidence versus execution-based similarity. We provide sensitivity analysis on the effect of varying $\lambda$ on execution accuracy in Appendix C.

### 3.2.3 MARGINAL PROBABILITY CALCULATION

In tasks where a language model (LM) is used to score multiple candidate answers for a given prompt, a common approach is to directly compute the conditional probability $P(X \mid Q, D)$, where $X$ is a candidate answer, $Q$ is the question, and $D$ represents few-shot demonstrations. However, this approach suffers from instability due to sensitivity to prompt formatting and content, a phenomenon we refer to as prompt sensitivity bias. Slight variations in the prompt (e.g., changes in demonstrations or formatting) can result in significant fluctuations in the predicted probabilities, making fair comparison across candidates difficult.

To illustrate this issue, consider candidate answers $A$ and $B$, question $Q$, and a set of demonstrations $D_1, D_2, \ldots, D_n$. The model's output probabilities may vary such that for some $i$, $P(A \mid D_i, Q) > P(B \mid D_i, Q)$, while for others, $P(A \mid D_i, Q) < P(B \mid D_i, Q)$. This inconsistency becomes especially problematic when the model assigns similar probabilities to both candidates, amplifying the unreliability of direct scoring.

To mitigate prompt sensitivity bias, we propose an alternative probability formulation based on marginalization. Rather than computing $P(X \mid Q, D_i)$ directly, we estimate the joint probability $P(X, Q)$, which is proportional to $P(X \mid Q)$ given that $P(Q)$ remains constant across all candidates. This is done by marginalizing over a set of sampled demonstrations:

$$P(X, Q) \approx \sum_{i=1}^{n} P(X, Q, D_i) \tag{6}$$

Due to the infeasibility of enumerating all possible demonstrations, we approximate the marginal by summing over a finite subset $D_1, D_2, \ldots, D_n$. The larger the sample size $n$, the more accurate the approximation of the marginal probability becomes.

A naive approach to estimating $P(X, Q, D_i)$ is to compute the product of the token-level probabilities predicted by the language model considering token probabilities are independent. However, this introduces a bias toward shorter sequences due to exponential decay in probabilities. To counteract this, a length-normalized formulation is necessary. A simple length penalty that divides the joint log-probability by the total number of tokens in the sequence (e.g., $\mid X \mid + \mid Q \mid + \mid D_i \mid$) does not sufficiently emphasize the probability of the candidate $X$, particularly when $Q$ and $D_i$ are long.

To address this, we apply a structured length penalty that ensures the generated candidate's likelihood retains appropriate emphasis. Let $r$, $s$, and $t$ denote the number of tokens in $X$, $Q$, and $D$ respectively, and let $x_j$, $q_k$, and $d_\ell$ represent the individual tokens of each component. We define:

$$P(X, Q, D) \approx \left( \prod_{j=1}^{r} P(x_j) \right)^{\frac{1}{r}} \cdot \left( \prod_{k=1}^{s} P(q_k) \cdot \prod_{\ell=1}^{t} P(d_\ell) \right)^{\frac{1}{s+t}} \tag{7}$$

This formulation applies separate length penalties to the candidate tokens and the prompt context, thereby amplifying the influence of the candidate answer in the joint probability.

## 4 EXPERIMENTS

### 4.1 EXPERIMENTAL SETUP

**Datasets.** We conduct our experiments on the development sets of two widely-used text-to-SQL benchmarks: BIRD (Li et al., 2024c) and SPIDER v1.0 (Yu et al., 2018). The BIRD dataset is designed to evaluate the generalizability of models in text-to-SQL. It features a diverse range of database schemas and question types, making it a strong benchmark for assessing semantic parsing capabilities in realistic scenarios. SPIDER v1.0 is a large-scale, cross-domain semantic parsing dataset that focuses on compositional generalization. Its development set contains complex natural language questions over unseen database schemas, and it has become the de-facto standard for evaluating text-to-SQL models in both academic and industrial settings.

**Metrics.** We evaluate model performance using execution accuracy, which measures the percentage of generated SQL queries that yield the same execution result as the ground truth query. Unlike syntactic accuracy metrics, execution accuracy reflects the semantic correctness of the output and is therefore more robust to superficial differences in SQL structure that do not affect the query result.

**Implementation Details.** To construct prompts for few-shot learning, we use a retrieval-augmented generation (RAG) method based on the approach described in the CodeS paper (Li et al., 2024b). For each input question, we retrieve five few-shot exemplars from a demonstration pool using the RAG method. We operate in a one-shot setting, where each of the five retrieved exemplars is inserted individually into the prompt to generate five distinct prompt variants. These prompt variants

are used to calculate the probability of generated candidate queries, allowing us to compute the marginal probability of each candidate by averaging its probabilities across different few-shot contexts. The prompt format follows the same structure as introduced in the CodeS paper (Li et al., 2024b) to maintain consistency with prior work. We use top-k sampling for candidate generation and retain 32 distinct queries per input question. This design provides a balance between diversity and computational efficiency while ensuring a sufficient candidate pool for downstream scoring.

**Baselines.** Our experiments include a range of competitive baselines across both model types and ensemble strategies. We evaluate Qwen-Coder-Instruct (Hui et al., 2024) models of various sizes to assess how our method performs on pre-trained models with different capacities. We also include CodeS (Li et al., 2024b) fine-tuned models, which are based on the GPT-2 (Radford et al., 2019) architecture and trained specifically for text-to-SQL generation, providing a strong task-specific benchmark. In terms of ensemble strategies, we compare our ensemble method against self-consistency (majority voting), MBR, and MBMBR (Jinnai et al., 2023).

## 4.2 RESULTS AND ANALYSIS

**Main Results.** Table 1 presents the execution accuracy of four fully unsupervised ensemble methods, Voting, MBR (Shi et al., 2022), MBMBR (Jinnai et al., 2023), and our proposed approach, evaluated across multiple fine-tuned and pre-trained models on the BIRD and SPIDER datasets. The results indicate that our method consistently achieves the highest accuracy across all model–dataset combinations, outperforming existing heuristic-based ensemble techniques. While MBMBR generally yields higher accuracy than both Voting and MBR, its performance occasionally falls below these simpler methods, highlighting potential instability. In contrast, Voting and MBR exhibit similar results, with only marginal differences across settings. These findings demonstrate the robustness and effectiveness of our method in enhancing execution accuracy on both datasets. To further demonstrate the effectiveness of our method in addressing the shortcomings of MBMBR, we have also provided our qualitative analysis and case study in Appendix D

Table 1: Main results with different models on BIRD (**B**) and SPIDER (**S**) development sets. We are Comparing our method with previous heuristic-based ensemble methods including voting, MBR, and MBMBR. ‡ and † shows statistical significance with p-value $< 0.01$ and $0.05$ respectively.

| Model | Ensemble method | B | S |
|---|---|---|---|
| SFTCodeS-7b (Li et al., 2024b) | Voting | 57.76 | 81.8 |
| | MBR (Shi et al., 2022) | 57.43 | 81.6 |
| | MBMBR (Jinnai et al., 2023) | 57.63 | 81.5 |
| | Ours | **58.21**$^†$ | **83.2**$^‡$ |
| SFTCodeS-15b (Li et al., 2024b) | Voting | 57.82 | 81.4 |
| | MBR (Shi et al., 2022) | 57.82 | 81.6 |
| | MBMBR (Jinnai et al., 2023) | 58.47 | 81.5 |
| | Ours | **59.45**$^‡$ | **82.5**$^†$ |
| Qwen2.5-Coder-7B-Instruct (Hui et al., 2024) | Voting | 58.41 | 76.1 |
| | MBR (Shi et al., 2022) | 57.76 | 76.2 |
| | MBMBR (Jinnai et al., 2023) | 58.54 | 77.1 |
| | Ours | **60.43**$^‡$ | **78.9**$^†$ |
| Qwen2.5-Coder-14B-Instruct (Hui et al., 2024) | Voting | 63.43 | 82.7 |
| | MBR (Shi et al., 2022) | 63.23 | 83.0 |
| | MBMBR (Jinnai et al., 2023) | 63.75 | 83.5 |
| | Ours | **64.08** | **83.7** |
| Qwen2.5-Coder-32B-Instruct (Hui et al., 2024) | Voting | 63.56 | 84.3 |
| | MBR (Shi et al., 2022) | 63.75 | 84.3 |
| | MBMBR (Jinnai et al., 2023) | 63.82 | 84.0 |
| | Ours | **64.08** | **84.8** |

**Ablation.** Table 2 presents the ablation study conducted on three representative models (SFTCodeS-7b, SFTCodeS-15b, and Qwen2.5-Coder-7B-Instruct) across the BIRD and SPIDER datasets. We begin by reporting the baseline MBMBR (Jinnai et al., 2023) performance. The "+marginal" row

replaces MBMBR's probability computation with our marginal probability calculation while retaining MBMBR's original scoring function, isolating the contribution of our probability estimation approach. The "+our scoring function" row instead applies our scoring method without marginal probability computation, allowing us to assess the effect of the scoring strategy independently. Finally, the "both (ours)" row applies both components, our scoring function and marginal probability calculation, representing the full version of our method.

Table 2 highlights several key findings from our ablation study. First, our heuristic scoring consistently outperforms MBMBR across all models and datasets, despite relying on the same probability estimates, demonstrating the effectiveness of the revised scoring strategy. Second, replacing MBMBR's probabilities with our marginal probability estimates yields comparable performance overall, with only minor improvements, except in the case of Qwen-7B on BIRD, where the marginal probabilities provide nearly a 1% gain. Finally, when the marginal probability calculation is combined with our scoring function, the performance improvements are consistent and more substantial across all settings. This effect arises because our scoring formulation more directly integrates candidate probabilities, allowing improvements in probability estimation to translate into greater performance gains. In contrast, MBMBR's reliance on probabilities is limited, such that even improved estimates cannot fully compensate for the shortcomings of its scoring formulation.

Table 2: Ablation study results with three different models on BIRD and SPIDER datasets investigating the effect of our proposed marginal probability calculation (+marginal) and our scoring function (+our scoring function) individually, and the effect of combining them together (+both).

| Ensemble Mehtod | SFTCodeS-7b | | SFTCodeS-15b | | Qwen-7B | |
| --- | --- | --- | --- | --- | --- | --- |
| | BIRD | SPIDER | BIRD | SPIDER | BIRD | SPIDER |
| MBMBR | 57.63 | 81.5 | 58.47 | 81.5 | 58.54 | 77.1 |
| +marginal | 57.63 | 81.5 | 58.34 | 81.6 | 58.74 | 77.1 |
| +our scoring function | 57.69 | 82.5 | 59.19 | 81.9 | 59.84 | 78.7 |
| +both (ours) | **58.21** | **83.2** | **59.45** | **82.5** | **60.43** | **78.9** |

**Analysis Across Difficulty Levels.** Figure 1 illustrates the performance of the Qwen-7B model on the BIRD and SPIDER datasets, analyzed across question difficulty levels and query lengths. While the dataset-defined difficulty levels are provided by BIRD and SPIDER, the query lengths are demonstrated in Fig. 1 by binning the ground-truth query length distribution into five intervals, and we report the performance of the baseline models for each bin. Results for models beyond Qwen-7B are presented in the Appendix E. The figure shows that our method consistently improves performance across all difficulty levels and query lengths, rather than being confined to a particular subset of questions. This indicates that the effectiveness of our approach is broadly applicable, making it beneficial for handling diverse text-to-SQL queries of varying complexity and length.

**Analysis Across Different Number of Candidates.** Table 3 presents the execution accuracy of Qwen models with 7B, 14B, and 32B parameters on both datasets, evaluated as a function of the number of generated candidates. In nearly all cases, accuracy increases with the number of candidates, although the rate of improvement diminishes as the number grows, eventually converging to the model's upper performance bound. When only a single candidate is available, all ensemble methods yield identical results, as no ensemble effect is possible. Conversely, with an infinite number of candidates, the distributions become exhaustive, and all methods converge to similar performance, since every possible execution result is represented. The key distinction emerges when the candidate pool is limited. Under these conditions, our method consistently outperforms competing baselines, as it more effectively incorporates candidate probabilities when the candidates distribution does not fully capture the underlying probability space. In table 3, † sign in each column indicates where our method achieves the most pronounced improvements over baselines across different number of candidates. Notably, larger models reach convergence with fewer candidates, reflecting their higher inherent capacity. Additional analyses for CodeS models are provided in the Appendix E.

**Analysis of Best Number of Candidates.** Building on the observations from table 3, Fig. 2 identifies the number of candidates at which the performance gap between our method and baseline approaches is maximized, referred to as the "best # candids." This value reflects the point at which our method yields the greatest advantage before convergence diminishes differences among ensemble methods. The figure reports results for both Qwen and CodeS models across the two datasets

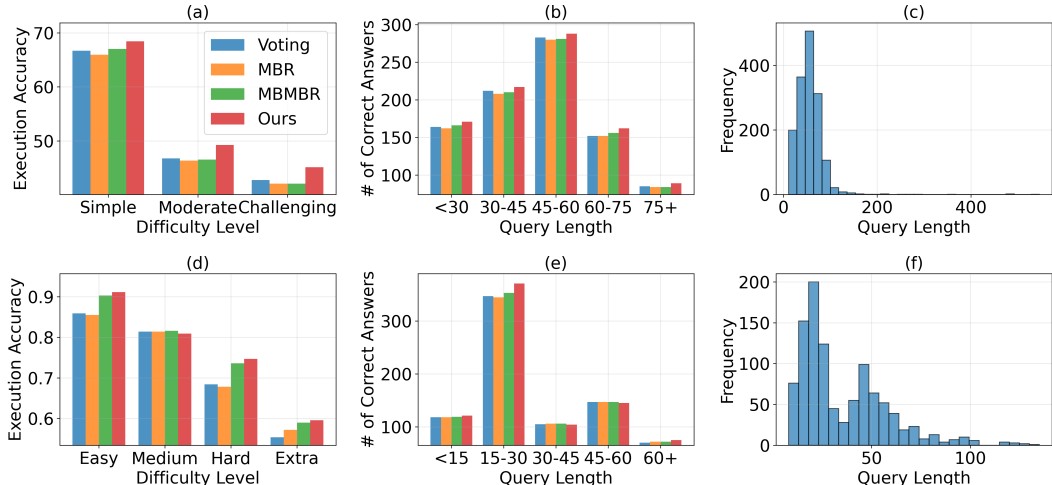

Figure 1: Performance analysis of Qwen-7b with heuristic-based ensemble methods across different question difficulty levels (subfigures (a) and (d)), and across different ground truth query lengths (subfigures (b) and (e)). The length distribution of the ground truth queries is also shown in subfigures (c) and (f). Subfigures (a), (b), and (c) are on BIRD while (d), (e), (f) are on SPIDER

Table 3: Performance analysis of Qwen models on BIRD and SPIDER when different number of candidates are available. The best accuracy across all baselines is shown in bold. † shows the accuracy that has the highest performance gap with other baselines across different number of candidates.

| # Candids | Ensemble Method | Qwen-7b | | Qwen-14b | | Qwen-32b | |
|---|---|---|---|---|---|---|---|
| | | BIRD | SPIDER | BIRD | SPIDER | BIRD | SPIDER |
| 2 | Voting | 41.26 | 65.4 | 51.56 | 74.1 | 56.58 | 76.4 |
| | MBR | 41.66 | 65.9 | 51.83 | 73.4 | 56.19 | 76.0 |
| | MBMBR | 42.31 | 68.7 | **52.93** | 74.7 | 56.84 | 78.8 |
| | Ours | **42.63** | **69.3** | 52.8 | **75.0** | **57.17** | **79.4**$^\dagger$ |
| 4 | Voting | 49.93 | 70.9 | 57.82 | 78.1 | 60.04 | 82.4 |
| | MBR | 49.87 | 71.8 | 57.69 | 77.9 | 59.97 | 82.1 |
| | MBMBR | 50.72 | 72.2 | **58.93** | 79.3 | 60.04 | 83.1 |
| | Ours | **51.04** | **73.1** | 58.8 | **79.9** | **60.63**$^\dagger$ | **83.2** |
| 8 | Voting | 54.95 | 73.1 | 60.76 | 80.5 | 62.78 | 83.9 |
| | MBR | 54.82 | 73.4 | 61.15 | 80.6 | 62.45 | 83.9 |
| | MBMBR | **55.48** | **74.0** | 61.8 | 80.7 | **62.84** | **84.1** |
| | Ours | 55.02 | 73.8 | **62.26** | **81.4**$^\dagger$ | 62.58 | 83.8 |
| 16 | Voting | 56.39 | 75.0 | 62.65 | 82.6 | 62.45 | 83.9 |
| | MBR | 56.58 | 75.0 | 62.52 | 82.7 | 62.71 | 83.6 |
| | MBMBR | 57.56 | 74.7 | 62.71 | 82.9 | 62.91 | 83.2 |
| | Ours | **57.95** | **75.5** | **63.49**$^\dagger$ | **83.1** | **63.04** | **84.4** |
| 32 | Voting | 58.41 | 76.1 | 63.43 | 82.7 | 63.56 | 84.3 |
| | MBR | 57.76 | 76.2 | 63.23 | 83.0 | 63.75 | 84.3 |
| | MBMBR | 58.54 | 77.1 | 63.75 | 83.5 | 63.82 | 84.0 |
| | Ours | **60.43**$^\dagger$ | **78.9**$^\dagger$ | **64.08** | **83.7** | **64.08** | **84.8** |

and across different model sizes. The results show a clear trend: as model size increases, the "best # candids" decreases. In other words, larger models require fewer candidates to achieve both higher overall performance and more accurate representation of the underlying candidate probability distribution. This finding highlights the efficiency of larger models in candidate utilization and further underscores the robustness of our method in improving performance across model scales.

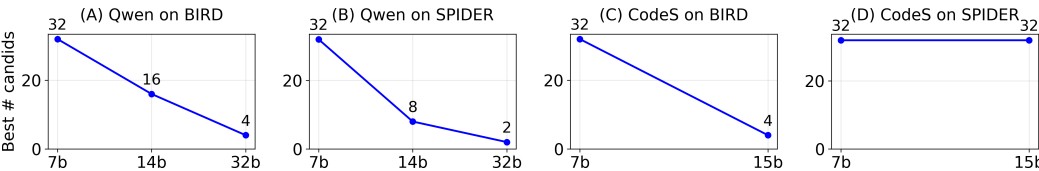

Figure 2: "Best # candids" (Y axis) with respect to model size (X axis).

**Cost-Performance Trade-off.** A key consideration in ensemble-based text-to-SQL systems is the trade-off between computational cost and the corresponding performance gains. Here, we compare the computational characteristics of the standard baselines (*Voting*, *MBR*, and *MBMBR*) with those of our proposed method. All three baselines follow the same computational pipeline until a final answer is selected: (1) *Candidate generation:* Each of the $n$ candidates is generated via one LLM call (MBMBR additionally obtains the probability during this step). (2) *Execution:* Each candidate is executed to obtain its resulting SQL output. (3) *Scoring:* All pairs of execution outputs are compared to compute their pairwise similarity.

Let $t$ denote the average LLM response time, $q$ the average execution time per candidate, and $e$ the time required to compare two execution results. The total response time for these baseline methods is: $T_{\text{baseline}} = n.t + n.q + \binom{n}{2}e$. On the other hand, our proposed method introduces an additional step to compute the marginal probability of each candidate using $p$ prompts. Since the initial LLM call provides one probability estimate, only $p-1$ additional prompts per candidate are needed. The resulting response time is therefore: $T_{\text{ours}} = n.p.t + n.q + \binom{n}{2}e$.

If we ignore the execution time and comparison time (assuming $q \approx 0$ and $e \approx 0$), the response time of our method is at most $p$ times larger than that of the baselines. However, in practice, this upper bound is seldom reached, especially when using smaller LLMs. Smaller models tend to produce lower-quality candidates, which significantly increases the execution time because many candidates fail or time out during execution. In such cases, the LLM response time $t$ becomes negligible compared to the execution cost $q$, and the overall response time of our method becomes comparable to that of the baselines despite requiring extra LLM calls.

Overall, the response time of our full method lies between **lower bound** equal to the baseline response time (when execution dominates), and an **upper bound** up to $p$ times the baseline response time (when LLM latency dominates). This analysis demonstrates that the additional probabilistic refinement introduced by our method does not necessarily result in a proportional increase in wall-clock time, particularly in settings where execution cost is the primary bottleneck.

## 5 CONCLUSION

We presented a new heuristic-based ensemble method for the task of NL2SQL that addresses two key weaknesses of existing approaches: (1) prompt sensitive probability calculation and (2) allowing candidates with low-probabilities to dominate the selection process. By introducing a new scoring function, our method leverages both pairwise utility and the intrinsic probability of each candidate, effectively reducing the risk of selecting implausible queries. Additionally, our marginal probability calculation over multiple prompt variations improves robustness against prompt-specific biases.

Extensive experiments on two datasets demonstrate that our method consistently outperforms state-of-the-art heuristic based ensemble techniques in execution accuracy, across both fine-tuned and pre-trained LLM settings. The ablation studies confirm that each proposed component contributes to the overall performance gains, with the best results achieved when both are applied together.

Future work may explore extending our approach to other structured prediction tasks beyond text-to-SQL (eg. other types of code generation), as well as investigating automatic prompt modification strategies to further enhance robustness in generated answer probability calculation. Additionally, evaluating our method against fine-tuned sample selectors constitutes promising future work, particularly for comparing in-distribution and out-of-distribution generalization

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

## A  LLM USAGE

In the preparation of this manuscript, large language models (LLMs) were employed as an auxiliary tool to improve the clarity and presentation of the text. Specifically, LLMs were used to polish the writing by correcting typographical errors, grammatical mistakes, and other surface-level issues. They were also used to compress longer passages into more concise formulations while preserving their technical meaning. In addition, LLMs assisted in the literature review process by suggesting potentially relevant related work, which was subsequently verified and curated by the authors to ensure comprehensive coverage and accuracy.

## B  DERIVATION DETAILS

We start form the following formula and start deriving the simplified version of the formula in Eq. 4

$$h^* = \arg\max_{h \in \mathcal{H}} \left( \mathbb{E}_{x,y \sim \mathcal{H}}[u(x,y)] - \mathbb{E}_{x,y \sim \mathcal{H} \setminus \{h\}}[u(x,y)] \right)$$

We further expand the expectations:

$$h^* = \arg\max_{h \in \mathcal{H}} \left( \sum_{x,y \in \mathcal{H}} u(x,y) \cdot P(x,y) - \sum_{x,y \in \mathcal{H} \setminus \{h\}} u(x,y) \cdot P(x,y) \right)$$

By assuming $x$ and $y$ are independent ($P(x,y) = P(x) \cdot P(y)$), we can further expand the equation to:

$$h^* = \arg\max_{h \in \mathcal{H}} \left( \sum_{x \in \mathcal{H}} P(x) \sum_{y \in \mathcal{H}} u(x,y) \cdot P(y) - \sum_{x \in \mathcal{H} \setminus \{h\}} P(x) \sum_{y \in \mathcal{H} \setminus \{h\}} u(x,y) \cdot P(y) \right)$$

After simplification we will reach to the following formula:

$$h^* = \arg\max_{h \in \mathcal{H}} \left( 2 \cdot P(h) \sum_{y \in \mathcal{H}} u(h,y) \cdot P(y) - P(h)^2 \cdot u(h,h) \right)$$

By defining $u(h,h) = 1$, we will have:

$$h^* = \arg\max_{h \in \mathcal{H}} \left( 2 \cdot P(h) \sum_{y \in \mathcal{H}} u(h,y) \cdot P(y) - P(h)^2 \right)$$

We can divide everything inside the $\arg\max$ my 2:

$$h^* = \arg\max_{h \in \mathcal{H}} \left( P(h) \sum_{y \in \mathcal{H}} u(h,y) \cdot P(y) - \frac{1}{2} \cdot P(h)^2 \right)$$

## C  SENSITIVITY ANALYSIS

Table 4 provides a sensitivity analysis of our only tunable hyper-parameter $\lambda$ using the execution accuracy of Qwen-7B on the BIRD benchmark. The following table compares MBMBR and our method across different values of $\lambda$.

The results highlight several key observations regarding the behavior of both methods across different values of the hyperparameter $\lambda$. First, MBMBR is largely insensitive to $\lambda$. Its execution accuracy

Table 4: Sensitivity analysis for hyper-parameter $\lambda$

| $\lambda$ | 0.1 | 0.2 | 0.3 | 0.4 | 0.5 | 0.6 | 0.7 | 0.8 | 0.9 |
|---|---|---|---|---|---|---|---|---|---|
| **MBMBR** | 58.47 | 58.47 | 58.47 | 58.47 | **58.54** | 58.47 | 58.47 | 58.47 | 58.47 |
| **Our Method** | **60.43** | 60.30 | 60.37 | 60.23 | 60.10 | 59.91 | 59.84 | 59.78 | 59.52 |

remains nearly constant over a wide range of values, showing minimal improvement as $\lambda$ changes. Its best performance within the examined range occurs at $\lambda = 0.5$, which is the value we report in the main results table for fairness.

In contrast, our method exhibits a smooth and predictable dependence on $\lambda$. Decreasing $\lambda$, which increases the influence of probabilities in the final score, slightly improves execution accuracy. This improvement occurs because the probabilistic component contributes more effectively, and the use of marginal probabilities yields more stable probability estimates.

Most importantly, our method consistently and substantially outperforms MBMBR across the entire range of $\lambda$. Taken together, these results demonstrate that the hyperparameter $\lambda$ is not overfitting to the test set. Instead, $\lambda$ provides a principled mechanism for adjusting the balance between utility-based similarity scoring and probability-aware scoring. Because our method produces more reliable probability estimates, reducing $\lambda$ naturally allows the probabilistic component to contribute more effectively, resulting in smooth and stable performance gains.

## D  QUALITATIVE ANALYSIS AND CASE STUDY

To illustrate the practical differences between our method and MBMBR, we present a representative case from the BIRD dataset. The following example shows candidate SQL queries generated by the QWEN-7B model.

**Question.**  *"List out the accounts who have the earliest trading date in 1995."*

**Query selected by our method.**

```
SELECT account_id
FROM trans
WHERE date LIKE '1995%'
ORDER BY date ASC
LIMIT 1;
```

**Query selected by MBMBR.**

```
SELECT DISTINCT account_id
FROM trans
WHERE date BETWEEN '1995-01-01' AND '1995-12-31'
ORDER BY date ASC
LIMIT 1;
```

**Analysis.**  The query selected by MBMBR is incorrect because the use of `DISTINCT` prevents reliable identification of the true earliest transaction when an account appears multiple times in 1995. In the candidate set provided by the model (32 candidates total), 12 queries contain this incorrect `DISTINCT` pattern, while only 8 candidates correspond to fully correct solutions. Although both patterns appear frequently, the correct candidates receive higher probability from the base model. Our method leverages these probability differences effectively through marginal-probability integration, allowing it to select the correct query. In contrast, MBMBR relies more on utility-based voting and weakly accounts for these probability signals, making it prone to selecting high-frequency but incorrect patterns such as `DISTINCT`.

This case study demonstrates how probability-aware scoring enables our method to avoid common structural errors and consistently identify higher-quality candidates.

# E   FURTHER ANALYSIS

For further analysis, we report the performance of SFTCodeS-7b (Figure 4) and SFTCodeS-15b (Figure 3) across varying question difficulty levels and different ground truth query lengths. Both figures demonstrate that our method consistently outperforms baselines across all levels of question difficulty as well as across queries of different lengths. This indicates that the observed improvements are not confined to a specific subset of questions but generalize across diverse levels of complexity and query characteristics. Together with the results from a third model (Figure 1), these findings confirm that our method exhibits robust performance across different types of questions and model scales.

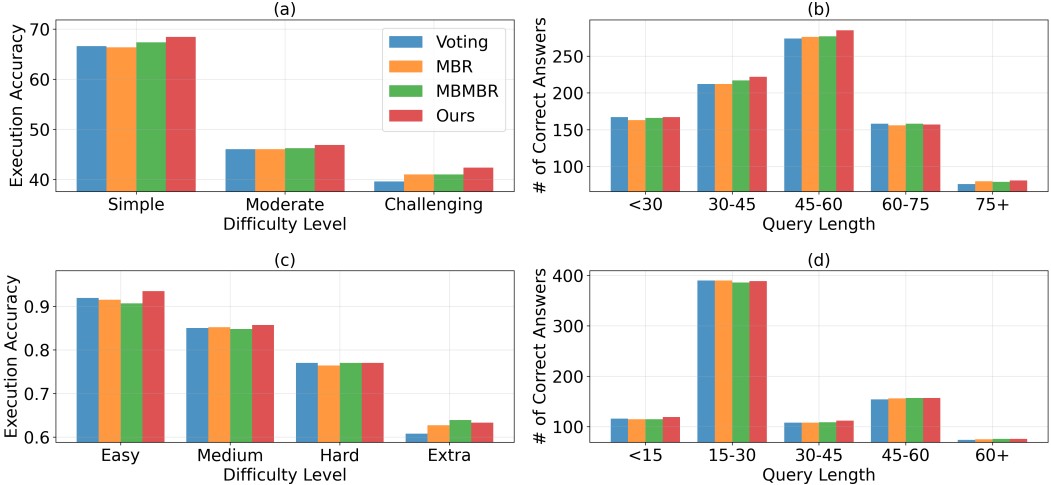

Figure 3: Performance analysis of ensemble methods across different question difficulty levels (subfigures (a) on BIRD and (c) on SPIDER), and across different ground truth query lengths (subfigures (b) on BIRD and (d) on SPIDER). Experiments are with SFTCodeS-15b.

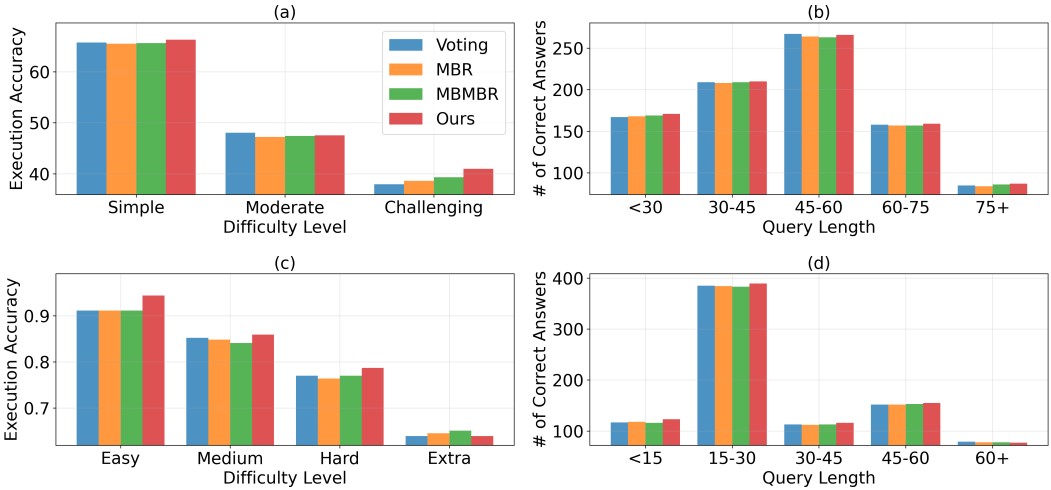

Figure 4: Performance analysis of ensemble methods across different question difficulty levels (subfigures (a) on BIRD and (c) on SPIDER), and across different ground truth query lengths (subfigures (b) on BIRD and (d) on SPIDER). Experiments are with SFTCodeS-7b.

We also report results on CodeS models with varying numbers of candidates in Table 5. The results exhibit the same pattern observed in Table 3, which presents results on Qwen models. Specifically, our approach outperforms the baselines in most cases across different candidate set sizes, further demonstrating the robustness and consistency of the method under varying conditions.

Table 5: Performance analysis of fine-tuned codeS models with different sizes on BIRD and SPIDER when different number of candidates are available. The best accuracy across all baselines is shown in bold. † shows the accuracy that has the highest performance gap with other baselines across different number of candidates.

| # Candids | Ensemble Method | SFTCodeS-7b | | SFTCodeS-15b | |
|---|---|---|---|---|---|
| | | BIRD | SPIDER | BIRD | SPIDER |
| 2 | Voting | 39.77 | 63.2 | 38.14 | 60.9 |
| | MBR | 39.77 | 63.2 | 38.27 | 60.4 |
| | MBMBR | 41.79 | 65.4 | 41.52 | **63.4** |
| | Ours | **41.85** | **66.2** | **42.50** | 63.3 |
| 4 | Voting | 48.31 | 72.5 | 45.57 | 69.8 |
| | MBR | 48.17 | 71.6 | 45.70 | 67.5 |
| | MBMBR | **49.61** | 72.5 | 47.26 | **70.4** |
| | Ours | 49.48 | **73.1** | 48.76† | **70.4** |
| 8 | Voting | 52.93 | 78.7 | 52.67 | **78.2** |
| | MBR | 52.93 | 78.7 | 52.41 | 77.0 |
| | MBMBR | 53.91 | 78.8 | 53.78 | 77.2 |
| | Ours | **54.17** | **79.1** | **54.43** | 77.6 |
| 16 | Voting | 55.80 | 80.4 | 56.58 | 80.2 |
| | MBR | 55.87 | 80.5 | 56.91 | 80.3 |
| | MBMBR | **56.13** | 80.7 | 56.98 | 80.2 |
| | Ours | 56.06 | **81.0** | **57.30** | **80.9** |
| 32 | Voting | 57.76 | 81.8 | 57.82 | 81.4 |
| | MBR | 57.43 | 81.6 | 57.82 | 81.6 |
| | MBMBR | 57.63 | 81.5 | 58.47 | 81.5 |
| | Ours | **58.21†** | **83.2†** | **59.45** | **82.5†** |

