# OpenReview forum: "Exploring Minimum Bayes Risk Decoding for Text-to-SQL Ensemble"
_ICLR.cc/2026/Conference — Submitted to ICLR 2026_

### Official Review · Reviewer_DCDh · 2025-10-30

**Soundness:** 3
**Presentation:** 4
**Contribution:** 3
**Rating:** 6
**Confidence:** 5

**Summary:**

LLMs have advanced in the field of text to SQL and many methods have leveraged ensembled approach. Existing heuristic based ensembles such as Minimum Bayes Risk (MBR) and Model Based MBR (MBMBR) either ignore model-predicted probabilities or allow low probability candidates to dominate the selection process, and they suffer from prompt sensitivity when estimating candidate likelihoods. They propose a novel heuristic-based ensemble method that directly incorporates each candidate’s own probability into its heuristic score while mitigating prompt sensitivity through marginal probability estimation across diverse prompts.

**Strengths:**

The paper identifies the limitations in the existing MBR and MBMBR, that they either ignore model predicted probabilities or allow low probability candidates to dominate the selection process, and they suffer from prompt sensitivity when estimating candidate likelihoods. They have proposed a heuristic based (these methods are efficient in comparison to supervised ensemble models and Prompt-based ensemble methods) ensemble method.

**Weaknesses:**

The proposed solution presents a practical refinement to the existing methods. The performance gains are consistent but relatively small. A deeper theoretical analysis could have also been helpful.

**Questions:**

How performance and runtime scale as the number of prompt variants and the number of generated candidates increase?

---

> ### Author Response · Authors · 2025-11-21
> **Response to Reviewer DCDh**
>
> We sincerely thank the reviewer for their valuable feedback and for helping us improve the quality of our paper. We also appreciate the positive assessments regarding the soundness, clarity of presentation, and contributions of our work. In particular, we are grateful for the recognition of our paper’s identification of key limitations in existing MBR and MBMBR methods, as well as the strengths of our proposed heuristic-based ensemble approach.
>
> ---
>
> ### **Response to Weaknesses**
>
> > *“The proposed solution presents a practical refinement to the existing methods. The performance gains are consistent but relatively small. A deeper theoretical analysis could have also been helpful.”*
>
> Thank you for this constructive comment. We agree that deeper analysis is important, and we appreciate the opportunity to clarify the significance of our improvements. As suggested, we have computed statistical significance values for the results presented in Table 1. For all fine-tuned models and the small pre-trained model, our improvements are statistically significant with *p* < 0.01 or *p* < 0.05. To further strengthen the contribution, we have additionally incorporated qualitative analyses in the appendix, which illustrate concrete examples of how our method improves candidate selection compared to existing ensemble strategies.
>
> ---
>
> ### **Response to Questions**
>
> > *“How performance and runtime scale as the number of prompt variants and the number of generated candidates increase?”*
>
> Thank you for raising this important question. In response, we have added a formal runtime analysis at the end of Section 4.2. We express the total latency of our method as a function of the number of prompt variants and candidate samples.
>
> Let
> - \( t \): average LLM response time
> - \( q \): average execution time per candidate
> - \( e \): time required to compare two execution results
> - \( p \): number of prompts used in marginal probability estimation
> - \( n \): number of candidates
>
> Then the total latency of our method is:
>
> $$
> T_{\text{ours}} = n \cdot p \cdot t + n \cdot q + \binom{n}{2} \cdot e.
> $$
>
> This formulation clarifies how each component of our pipeline scales and offers direct comparison to existing heuristic-based ensemble approaches. Additional discussion has been included in the updated manuscript.
>
> ---

---

### Official Review · Reviewer_N3kd · 2025-10-31

**Soundness:** 3
**Presentation:** 2
**Contribution:** 2
**Rating:** 4
**Confidence:** 3

**Summary:**

The paper proposes a improved version of Minimum Bayes Risk (MBR) decoding algorithm to address the text-to-SQL problem. The authors claim that the original MBR algorithm is ignorant to the candidate’s own probability, which can potentially lead to a reward hacking scenario where the candidate with highest utility but lowest likelihood is selected. The proposed method directly incorporates each candidate's own probability into its score and uses marginal probability estimation across diverse prompts to ensure stability and accuracy. The evaluation shows performance improvement on BIRD and SPIDER compared to traditional MBR and MBMBR methods.

**Strengths:**

- The paper is targeted on candidate selection of LLMs, which is a high-impact problem to solve. The paper is well motivated given the clear limitations from previous work.
- The overall mythology design is technically sound and reasonable.

**Weaknesses:**

- The absolute performance improvement in BIRD and SPIDER appears to be small, which makes the effectiveness of the proposed method on text-to-SQL (and potentially other tasks) questionable.
- It is not totally clear to the readers where the performance improvement is from.
- While being seemingly applicable to other language tasks where model ensemble is used, only text-to-SQL results are shown in this paper. Therefore, the generalizability of the methodology, while it may be exist theoretically, is not well evaluated in this paper.
- How hyperparameter is selected and how is impacts the performance is not well discussed in the paper.

**Questions:**

- Is it the desired behavior of MBR decoding that a low-probability candidate can be selected over a high-probability one when it has a higher expected utility due to its similarity to the other candidates in the sample pool? This seems to be the core idea of MBR to correct the shortcomings of relying solely on the model's assigned probability by seeking consensus.
- Can authors give concrete examples that in text-to-SQL benchmarks, the reward hacking described in section 3.1 actually happens and it can be mitigated by the proposed method. This brings greater confidence to readers that the performance improvement presented in Table 1 is indeed a consequence of addressing the reward hacking issue.
- The example in Section 3.1 seems to be mathematically incorrect. P(A) and P(B) cannot be 0.9 at the same time.
- How does the proposed method quantitatively compare to fine-tuned sample selectors, e.g. the ones used in CHASE-SQL or MSc-SQL? Is there any quantitative results showing the in-distribution and out-of-distribution accuracy supporting the argument on low generalizability in Section 1?
- How does the proposed method compare to other MBR extensions for LLM decoding, like [1] and [2].
- It appears that the proposed method can also be applied to other language tasks. Can authors show performance results on other tasks, like Q&A, code generation, translation, etc., so that its generalizability can also be validated?
- Can authors clarify what the statistical significance means in table 1? Does it mean its performance superiority over other methods in the same block is statistically significant with p? If so, what about the results of 14B and 32B models?

[1]. Daheim, Nico, et al. "Uncertainty-aware decoding with minimum bayes risk." *arXiv preprint arXiv:2503.05318* (2025).

[2]. Heineman, David, Yao Dou, and Wei Xu. "Improving minimum bayes risk decoding with multi-prompt." *Proceedings of the Conference on Empirical Methods in Natural Language Processing. Conference on Empirical Methods in Natural Language Processing*. Vol. 2024. 2024.

---

> ### Author Response · Authors · 2025-11-21
> **Response to Reviewer N3kd (Part 1)**
>
> We sincerely thank the reviewer for the time and effort spent evaluating our work and for providing thoughtful and constructive feedback that has helped us improve the quality of the paper.
>
> We also appreciate the positive remarks regarding our contributions. In particular, we thank the reviewer for acknowledging that our target problem (candidate selection for LLMs) is “a high-impact problem to solve,” and that “the overall methodology design is technically sound and reasonable.” We are grateful for this recognition and for the opportunity to further strengthen our work based on your comments.
>
> ---
>
> ## **Response to Weaknesses**
>
> > **W1. _“The absolute performance improvement in BIRD and SPIDER appears to be small, which makes the effectiveness of the proposed method on text-to-SQL (and potentially other tasks) questionable.”_**
>
> Thank you for this constructive comment. While the absolute improvement may appear modest at first glance, our proposed method *consistently* improves performance across **all** evaluated models and **all** datasets. Importantly, the gains are statistically significant for fine-tuned models and our small pre-trained model, with **p-values < 0.01 or 0.05**, demonstrating that these improvements are unlikely to be due to randomness. Given the maturity and competitiveness of text-to-SQL benchmarks such as BIRD and SPIDER, even small absolute gains are meaningful, particularly when achieved through a lightweight post-training decoding strategy that requires *no* additional model updates or task-specific supervision.
>
> ---
>
> > **W2. _“It is not totally clear to the readers where the performance improvement is from.”_**
>
> Thank you for raising this point. To address this, we included an ablation study (Table 2) that isolates the contributions of the two key components of our method: (1) incorporating each candidate's own probability into its score, and (2) estimating marginal probabilities across diverse prompts. The ablations show that **each component individually contributes to performance improvement**, and that **the two components are complementary**. When combined, they reinforce each other and yield even larger gains than either component alone, clarifying where the improvement originates and why the full method performs best.
>
> ---
>
> > **W3. _“While being seemingly applicable to other language tasks where model ensemble is used, only text-to-SQL results are shown in this paper. Therefore, the generalizability of the methodology, while it may exist theoretically, is not well evaluated in this paper.”_**
>
> Thank you for highlighting the broader applicability of our method and for recognizing its potential beyond the text-to-SQL domain. In this work, we focused our evaluation exclusively on text-to-SQL due to its structured nature and the availability of robust execution-based metrics that make it a strong testbed for candidate selection algorithms. Based on your helpful suggestion, we have updated the **Conclusion** section to explicitly state that exploring the generalizability of our algorithm to other tasks (such as code generation, question answering, and translation) is an important direction for future work.

---

> ### Author Response · Authors · 2025-11-21
> **Response to Reviewer N3kd (Part 2)**
>
> > **W4. _“How hyperparameter is selected and how it impacts the performance is not well discussed in the paper.”_**
>
> Based on your feedback we have added new content and analyses to clarify the roles, selection, and robustness of each hyperparameter in subsection 3.2.2:
>
> **1. Added a New Subsection on Hyperparameter Discussion (Sec. 3.2.2).**
> We now include a dedicated subsection explaining the purpose and behavior of each hyperparameter in the utility function. In particular, we clarify that the constants $e^{-2}$ and $e^{-1}$ serve as *fixed penalties* to enforce the ordering:
> (1) candidates with error < (2) partially valid candidates < (3) valid candidates.
> We further demonstrate that as long as this ordering is preserved, the exact values of these constants have negligible effect. For example, replacing them with $e^{-3}$ and $e^{-2}$ yields nearly identical outcomes, as the minimum score for valid queries remains  $e^{0}$.
>
> **2. Clarified the Motivation for the Threshold $\epsilon$.**
> We added an explanation showing that $\epsilon = 0.2$ is informed by the empirical distribution of `NULL` values in valid SQL queries from the training data. Approximately 20% `NULL` values is common among correct queries, while larger proportions typically indicate partially invalid structure. Hence, $\epsilon = 0.2$ provides a principled and stable boundary for distinguishing between partial and fully valid candidates. We use the same value across all models and datasets in all our experiments.
>
> **3. Added Sensitivity Analysis for the Tunable Hyperparameter $\lambda$.**
> We now include a sensitivity analysis evaluating $\lambda$, the only tunable hyperparameter in our method (Appendix C). Using execution accuracy on **BIRD** with **Qwen-7B**, we compare MBMBR and our method across a wide range of $\lambda$ values:
>
> | λ            | 0.1   | 0.2   | 0.3   | 0.4   | 0.5   | 0.6   | 0.7   | 0.8   | 0.9   |
> |--------------|-------|-------|-------|-------|-------|-------|-------|-------|-------|
> | **MBMBR**    | 58.47 | 58.47 | 58.47 | 58.47 | **58.54** | 58.47 | 58.47 | 58.47 | 58.47 |
> | **Our Method** | **60.43** | 60.30 | 60.37 | 60.23 | 60.10 | 59.91 | 59.84 | 59.78 | 59.52 |
>
> **Key observations:**
> - **MBMBR is largely insensitive to $\lambda$**: performance remains nearly constant, with its best value occurring at $\lambda = 0.5$, which we use for fairness in main-table comparisons.
> - **Our method exhibits smooth and predictable dependence on $\lambda$**: slightly decreasing $\lambda$ increases the influence of candidate probabilities, and when combined with marginal probability estimation, this results in higher stability and stronger execution accuracy.
> - **Our method consistently and substantially outperforms MBMBR across all tested values of $\lambda$**.
>
> These additions strengthen the clarity and transparency of our hyperparameter design and illustrate that the method is robust across a wide range of settings.
>
> ---
>
> ## **Response to Questions**
>
> > **Q1. _“Is it the desired behavior of MBR decoding that a low-probability candidate can be selected over a high-probability one…?”_**
>
> Classical MBR decoding assumes that all candidate probabilities are *uniform*, and therefore (even when the model provides useful probability signals) MBR cannot utilize them, as its scoring relies solely on pairwise utility (i.e., candidate similarity). MBMBR was introduced to address this limitation by incorporating probabilities into the MBR formulation. However, as our analysis shows, the way MBMBR integrates probabilities is not sufficiently effective and can still allow low-probability candidates to dominate due to reward hacking. This motivates our proposed formulation, which incorporates candidate probabilities in a more principled and impactful way.
>
> ---
>
> > **Q2. _“Can authors give concrete examples… that reward hacking actually happens and can be mitigated by the proposed method?”_**
>
> Following your feedback, we have included qualitative case studies in the appendix of the revised version (Appendix D). For convenience, we present one representative example here (candidates generated by Qwen-7B):
>
> **Question:**
> *“List out the accounts who have the earliest trading date in 1995.”*
>
> **Our method selects:**
> SELECT account_id
> FROM trans
> WHERE date LIKE '1995%'
> ORDER BY date ASC
> LIMIT 1;
>
> **MBMBR selects:**
> SELECT DISTINCT account_id
> FROM trans
> WHERE `date` BETWEEN '1995-01-01' AND '1995-12-31'
> ORDER BY `date` ASC
> LIMIT 1;
>
> The MBMBR-selected query is incorrect because the use of DISTINCT prevents the identification of the earliest transaction for accounts with multiple entries in 1995. Among the 32 generated candidates, 12 contain this incorrect DISTINCT pattern, while only 8 candidates are correct. Because correct candidates have higher model-assigned probabilities, our method successfully prioritizes them, mitigating this reward hacking behavior by properly incorporating probability into the ensemble decision

---

> ### Author Response · Authors · 2025-11-21
> **Response to Reviewer N3kd (Part 3)**
>
> > **Q3. “The example in Section 3.1 seems to be mathematically incorrect. P(A) and P(B) cannot be 0.9 at the same time.”**
>
> Thank you for catching this typo. We have corrected the numerical values in the revised version to ensure they form a valid probability distribution. We appreciate the reviewer’s careful reading.
>
> ---
>
> > **Q4. “How does the proposed method compare to fine-tuned sample selectors…?”**
>
> We appreciate this important question. Our current study focuses on comparing our method to training-free heuristic ensemble approaches, including MBR and MBMBR, rather than fine-tuned sample selectors such as those in CHASE-SQL or MSc-SQL. These learned selectors require substantial task-specific supervision and introduce training costs and distribution-shift challenges that differ from the scope of our decoding-level method.
>
> In response to your feedback, we have added a clarification in the conclusion section stating that evaluating our method against fine-tuned sample selectors constitutes promising future work, particularly for comparing in-distribution and out-of-distribution generalization under a unified evaluation framework.
>
> ---
>
> > **Q5. _“How does the proposed method compare to other MBR extensions for LLM decoding, like [1] and [2]?”_**
>
> > [1]. Daheim, Nico, et al. "Uncertainty-aware decoding with minimum bayes risk." arXiv preprint arXiv:2503.05318 (2025).
>
> >[2]. Heineman, David, Yao Dou, and Wei Xu. "Improving minimum bayes risk decoding with multi-prompt." Proceedings of the Conference on Empirical Methods in Natural Language Processing. Conference on Empirical Methods in Natural Language Processing. Vol. 2024. 2024.
>
> Thank you for this insightful question. We clarify below how our method relates to these two extensions and why they are complementary rather than overlapping.
>
> **Comparison with [1].**
> Daheim et al. [1] incorporate *model weight uncertainty* into MBR decoding by estimating the posterior distribution over model parameters. Each candidate is generated using a different set of sampled model weights, resulting in candidates with *different uncertainty values*, which are then used as an additional signal for ranking. This approach is particularly effective when ensembling multiple perturbed models or weight samples.
>
> Our method, in contrast, assumes a *single* set of model weights and a *single* model for candidate generation. Under this setting, all candidates naturally share the same model-weight uncertainty, making the uncertainty term used in [1] uninformative. Instead, we use the *predicted probability of each candidate* as the key signal incorporated into the MBR objective. Because these probabilities differ across candidates even under a fixed model, they provide a meaningful and complementary ranking signal.
>
> Importantly, in scenarios where candidates come from *multiple models* or *multiple weight samples*, our method can be directly combined with the technique in [1] by incorporating both candidate probability and candidate-specific uncertainty into the scoring function.
>
> **Comparison with [2].**
> Heineman et al. [2] show that generating candidates using *multiple diverse prompts* leads to a higher-quality candidate pool and improved MBR decoding. Their focus is on improving the *diversity* of the candidate set itself.
>
> Our method, however, generates candidates using **only a single prompt**. We use multiple prompts **solely for the purpose of marginal probability estimation**, not for candidate generation. This yields more robust and stable probability estimates while keeping the candidate pool consistent across methods.
>
> While orthogonal, these two approaches are compatible: one could adopt the multi-prompt candidate generation strategy from [2] and then apply our probability-based scoring, potentially benefiting from both improved diversity and better candidate ranking.
>
> **Summary.**
> Both [1] and [2] tackle aspects of MBR that are different from our goal. Their techniques improve *uncertainty modeling* or *candidate diversity*, whereas our approach improves the *scoring and ranking* mechanism through probability-aware marginal estimation. All three approaches can be combined when desired, and our method is complementary to both.

---

> ### Author Response · Authors · 2025-11-21
> **Response to Reviewer N3kd (Part 4)**
>
> > **Q6. _“It appears that the proposed method can also be applied to other language tasks. Can authors show performance results on other tasks, like Q&A, code generation, translation, etc., so that its generalizability can also be validated?”_**
>
> Thank you for this insightful question. As we mentioned in our response to **Weakness 3**, the current version of our paper focuses exclusively on the text-to-SQL task. While our method is, in principle, applicable to a broad range of language generation tasks, we chose to constrain the scope of this work to maintain experimental clarity and ensure a focused evaluation.
>
> To make this limitation explicit, we have added to the conclusion section to the revised paper, where we state that evaluating our algorithm on additional tasks such as code generation, Q&A, and translation is an important direction for future work. We appreciate the reviewer highlighting this point and agree that extending the evaluation to other tasks is a natural next step.
>
> ---
>
> > **Q7. _“Can authors clarify what the statistical significance means in Table 1? Does it mean its performance superiority over other methods in the same block is statistically significant with p? If so, what about the results of 14B and 32B models?”_**
>
> We thank the reviewer for the opportunity to clarify this. In Table 1, the statistical significance markers indicate that the performance improvement of our method over the **best-performing baseline in the same block** is statistically significant at **p < 0.01** or **p < 0.05**. This applies to the **small pre-trained models** and **fine-tuned models**, for which our method produces consistent and statistically significant gains.
>
> For larger models (14B and 32B), the effect is less pronounced. As model size increases, the base model becomes more confident and produces less diverse candidate outputs, making it inherently more difficult to achieve large or statistically significant improvements through candidate-selection strategies alone. We added clarifying text to the paper to reflect this phenomenon and explain why the statistical significance decreases as model size increases.

---

### Official Review · Reviewer_TztK · 2025-10-31

**Soundness:** 3
**Presentation:** 3
**Contribution:** 3
**Rating:** 4
**Confidence:** 5

**Summary:**

This paper addresses the task of Text-to-SQL, focusing on improving ensemble methods for candidate selection. The authors identify two key problems in existing heuristic-based ensembles like Minimum Bayes Risk (MBR) and Model-Based MBR (MBMBR): (1) they are sensitive to prompt formatting ("prompt sensitivity bias") when estimating candidate probabilities, and (2) their scoring can be dominated by low-probability candidates that are similar to others, as they don't sufficiently value a candidate's own probability. The paper proposes a novel heuristic-based method with two components to solve this. First, it introduces a new probabilistic scoring function that explicitly incorporates each candidate's own probability $P(h)$ into its utility score. Second, it proposes a "marginal probability calculation" to mitigate prompt sensitivity by estimating a candidate's probability as an average over multiple, diverse few-shot demonstrations. Experiments on the SPIDER and BIRD benchmarks show that the proposed method consistently outperforms standard baselines like voting, MBR, and MBMBR.

**Strengths:**

- **S1.** The paper identifies a clear and relevant set of problems with existing heuristic-based ensemble methods, namely the "prompt sensitivity bias" and the failure of standard MBR/MBMBR to properly incorporate a candidate's own probability, $P(h)$.
- **S2.** The proposed two-part solution is technically sound and well-motivated. The derivation of the new scoring function (Eq. 4), which provides a principled way to integrate $P(h)$ into the MBR framework, is a solid contribution. Furthermore, the idea of using marginal probabilities (Sec 3.2.2) to mitigate prompt sensitivity is an effective approach.
- **S3.** The experimental validation is thorough. The authors demonstrate the method's effectiveness across two standard benchmarks (BIRD, SPIDER) and, importantly, across multiple model families. The ablation study in Table 2 clearly isolates the individual contributions of the two components, showing they are complementary.

**Weaknesses:**

- **W1. Hyperparameter Complexity and Sensitivity:** The proposed utility function in Section 3.1.1 (Eq. 5) introduces a significant number of hard-coded, non-obvious hyperparameters (e.g., $e^{-2}$, $e^{-1}$, $\epsilon$). This design feels brittle and raises concerns about its practical applicability, as it seems to require extensive tuning. While the experiments show consistent gains, there is a strong possibility that these are the result of an "overfitted" set of hyperparameters specific to the (model, dataset) pairs tested. The paper lacks a sensitivity analysis for these hyperparameters, making it difficult to assess the method's general robustness.
- **W2. Unaddressed Cost-Benefit Trade-off:** A significant weakness of the paper is its failure to address the massive computational cost of the proposed probability estimation method (Sec 3.2.2) and its trade-off with performance. The method requires running $n$ (e.g., 5) distinct LLM calls per candidate to estimate its marginal probability. For the paper's setting of 32 candidates, this amounts to 160 LLM probability estimations to select a single answer. This is an orders-of-magnitude increase in cost compared to simple Self-Consistency (SC) Voting (which has 0 additional calls). The experimental results (Table 1) show performance gains of ~0.5-2% over these much cheaper baselines. For a real-time application like Text-to-SQL, this marginal gain seems to come at an unacceptable cost. The paper must provide a detailed analysis of this trade-off (e.g., Latency vs. Accuracy, Cost vs. Accuracy).
- **W3. Lack of Qualitative Case Studies:** The paper's motivation in Section 3.1 hinges on a key flaw of MBMBR—that it can select low-probability candidates. While an abstract example ({A, B, C}) is provided to illustrate this, the paper lacks a concrete, qualitative case study from the actual SPIDER or BIRD datasets. Showing a real example where MBMBR fails by picking an implausible-but-similar SQL query, and how the proposed scoring function (Eq. 4) corrects this error, would significantly strengthen the paper's motivation.
- **W4. Minor Presentation Error:** The illustrative example in Section 3.1 has a distracting error. The probabilities given ($P(A)=0.9$, $P(B)=0.9$, $P(C)=0$) sum to 1.8, not 1. This is a minor but sloppy mistake that should be corrected.

**Questions:**

- **Q1. Cost-Performance Trade-off:** My main concern is the practicality of the method. Could the authors provide a detailed trade-off analysis, comparing Latency/Cost versus Execution Accuracy for the proposed method against the cheaper baselines (SC-Voting, MBR, MBMBR)? Given the high cost of marginal probability estimation (Sec 3.2.2), how do the authors justify this trade-off for a real-time task?
- **Q2. Utility Function Motivation:** What is the theoretical or empirical motivation for the specific piecewise exponential design of the utility function in Eq. 5? Why this particular form over a simpler, linear penalty or a standard, non-piecewise function?
- **Q3. Utility Function Motivation and Sensitivity:** The utility function in Eq. 5 appears complex and relies on several hard-coded values. What is the theoretical or empirical motivation for its specific piecewise exponential design? Why were the penalties for empty/error results and results with high None values set specifically to $e^{-2}$ and $e^{-1}$, respectively? Could the authors provide a sensitivity analysis for the tunable hyperparameters in this function, namely the None threshold $\epsilon$ and the Jaccard similarity scaling parameter $\lambda$? This analysis is crucial to alleviate concerns about "overfitting" these hyperparameters to the test sets.
- **Q4. Qualitative Analysis:** To make the motivation in Section 3.1 more concrete, could the authors provide a qualitative case study from the BIRD or SPIDER dataset? Specifically, can you show a real example where MBMBR selects an incorrect, low-probability query (as per the {A, B, C} example), and demonstrate how your proposed scoring function (Eq. 4) successfully identifies the correct, higher-probability candidate?

---

> ### Author Response · Authors · 2025-11-21
> **Response to Reviewer TztK (Part 1)**
>
> We sincerely thank the reviewer for their valuable feedback and for taking the time to provide detailed comments that have helped us improve the quality of the paper.
>
> We also appreciate the reviewer’s positive remarks regarding our work. In particular, we thank you for noting that *“the paper identifies a clear and relevant set of problems”* and that *“the proposed two-part solution is technically sound and well-motivated.”* We are also grateful for your comments that *“the experimental validation is thorough”* and that *“the ablation study in Table 2 clearly isolates the individual contributions.”* Your recognition of these aspects is very encouraging.
>
> ---
>
> ## Response to Weakness W1: Hyperparameter Complexity and Sensitivity
>
> > **W1. Hyperparameter Complexity and Sensitivity:** The proposed utility function in Section 3.1.1 (Eq. 5) introduces a significant number of hard-coded, non-obvious hyperparameters (e.g., , , ). This design feels brittle and raises concerns about its practical applicability, as it seems to require extensive tuning. While the experiments show consistent gains, there is a strong possibility that these are the result of an "overfitted" set of hyperparameters specific to the (model, dataset) pairs tested. The paper lacks a sensitivity analysis for these hyperparameters, making it difficult to assess the method's general robustness.
>
> Thank you for this constructive and helpful comment. Based on your feedback, we added the discussion of hyperparameters and added new analyses to clarify their roles and robustness:
>
> **1. Added Subsection on Hyperparameter Discussion (Sec. 3.2.2).**
> In the revised version, we introduce a dedicated subsection that explains the purpose and behavior of each hyperparameter in the utility function. We clarify that the constants $e^{-2}$ and $e^{-1}$ are *fixed penalties* designed solely to enforce the ordering:
> (1) error candidates < (2) partially valid candidates < (3) fully valid candidates.
> Our experiments confirm that as long as this ordering is preserved, the exact numerical values have negligible impact. For example, replacing $e^{-2}$, $e^{-1}$ with $e^{-3}$, $e^{-2}$ yields nearly identical performance because the minimum score for valid candidates remains $e^{0}$.
>
> **2. Motivation for the $\epsilon$ Threshold.**
> We now clarify that $\epsilon = 0.2$ is derived from the empirical distribution of `NULL` values in ground-truth SQL queries in the training set. Approx. 20% `NULL` values is common among valid queries; exceeding this threshold typically reflects a partially invalid structure. Thus, $\epsilon = 0.2$ provides a principled boundary for distinguishing partial vs. valid results. We fixed $\epsilon$ across all of our experiments on all models and datasets.
>
> **3. Sensitivity Analysis for the Tunable Hyperparameter $\lambda$**
> To further address your concern, we added a sensitivity analysis evaluating our only tunable hyperparameter $\lambda$ in **Appendix C**.
> As requested, we provide a sensitivity analysis using the execution accuracy of **Qwen-7B** on the **BIRD** benchmark. The following table compares MBMBR and our method across different values of $\lambda$:
>
> | λ            | 0.1   | 0.2   | 0.3   | 0.4   | 0.5   | 0.6   | 0.7   | 0.8   | 0.9   |
> |--------------|-------|-------|-------|-------|-------|-------|-------|-------|-------|
> | **MBMBR**    | 58.47 | 58.47 | 58.47 | 58.47 | **58.54** | 58.47 | 58.47 | 58.47 | 58.47 |
> | **Our Method** | **60.43** | 60.30 | 60.37 | 60.23 | 60.10 | 59.91 | 59.84 | 59.78 | 59.52 |
>
> **Key observations:**:
> - MBMBR is relatively insensitive to λ: Execution accuracy cannot increase across a wide range of values and it stays almost the same. Its best performance in the above range occurs at $\lambda = 0.5$, which is the value we reported in the main result table for fairness.
> - Our method shows smooth, predictable dependence on λ: increasing the effect of probabilities (decreasing $\lambda$) slightly will result in a better execution accuracy since the probabilities are contributing to the final score more effectively and the use of marginal probability makes the probabilities more stable.
> - **Our method substantially outperforms MBMBR** across the full range of $\lambda$.
>
> Taken together, these results demonstrate that **the hyperparameter $\lambda$ is not overfitting the test set**. Rather, it provides a mechanism for adjusting the balance between utility-based similarity scoring and probability-aware scoring. Because our method achieves more reliable probability estimates, reducing $\lambda$ naturally allows the probabilistic component to “shine” and improves performance in a smooth and stable manner.
>
> We hope this detailed explanation and the quantitative sensitivity analysis sufficiently address the reviewer’s concern.

---

> ### Author Response · Authors · 2025-11-21
> **Response to Reviewer TztK (Part 2)**
>
> ## Response to Weakness W2: Unaddressed Cost–Benefit Trade-off
>
> > **W2. Unaddressed Cost-Benefit Trade-off:**
>
> Thank you for highlighting this important issue. Based on your helpful feedback, we have added a detailed **Cost–Performance Trade-off** subsection at the end of Section 4.2 in the revised paper.
>
> All three baselines follow the same computational pipeline until a final answer is selected:
> 1.  **Candidate generation**: Each of the $n$ candidates is generated via one LLM call (MBMBR additionally obtains the probability during this step).
> 2. **Execution**: Each candidate is executed to obtain its resulting SQL output.
> 3. **Scoring**: All pairs of execution outputs are compared to compute their pairwise similarity.
>
> To formalize the latency of the proposed method relative to standard baselines, we define:
>
> - \( t \): average LLM response time
> - \( q \): average execution time per candidate (SQL execution time)
> - \( e \): time required to compare two execution results
> - \( p \): number of prompts used in marginal probability estimation
> - \( n \): number of candidates
>
> The total latency of standard heuristic-based **baselines** (Voting, MBR, MBMBR) is:
>
> $
> T_{\text{baseline}} = n \cdot t + n \cdot q + \binom{n}{2} \cdot e.
> $
>
> **Our method** modifies only the probability-estimation component, replacing the single LLM call per candidate with \(p\) calls:
>
> $
> T_{\text{ours}} = n \cdot p \cdot t + n \cdot q + \binom{n}{2} \cdot e.
> $
>
> ### Practical Trade-off Behavior
>
> The relative overhead depends primarily on the relationship between \(t\) (LLM inference time) and \(q\) (execution time):
>
> - **Small or mid-size models (where $t \ll q$)**
>   In this regime, SQL execution dominates total latency. The additional LLM calls introduce negligible cost, and the runtime of our method is effectively identical to the baselines. This makes the approach fully practical for real-time scenarios using efficient LLMs.
>
> - **Large LLMs (where $t \gg q$)**
>   Here, LLM inference becomes the bottleneck. In the worst case, our latency increases by a factor of \(p\):
>   $
>   T_{\text{ours}} \approx p \cdot T_{\text{baseline}}.
>   $
>   However, we emphasize that **\(p\) is fully controllable**. In practice, we observe that even small values of \(p\) (e.g., 2–3) substantially reduce prompt-sensitivity variance while maintaining most of the accuracy gains. Thus, users can directly tune \(p\) to trade off cost and performance based on their deployment constraints. Additionally, the extra LLM calls for the probability estimation can be **executed in parallel**. So with enough computational resources and parallelization, The **response time can stay the same** as baseline methods.
>
> ### Summary
>
> Our method has almost **no latency drawback for smaller models**, which are commonly used in real-time systems. For larger models, practitioners may **reduce \(p\)** to achieve the desired balance between stability and latency. In addition to that, they can use **parallelization** to achieve the same latency as the baselines for the larger models.
>
> > ... This is an orders-of-magnitude increase in cost ...
>
> Thank you for expressing your concern and we are more than happy to clarify that the actual cost of our method falls between:
> - a **lower bound** equal to baseline cost (execution-dominated regimes), and
> - an **upper bound** of at most \(p\)× baseline cost (LLM-dominated regimes). ($p=5$ in the paper)
>
> This expanded discussion now clearly articulates the scenarios in which our method incurs additional cost and when it does not, thereby addressing the practical trade-offs you raised. We thank the reviewer again for pointing out this important aspect, which helped us improve the clarity and completeness of our analysis.
>
> ---
>
> ## Response to Weakness W3: Lack of Qualitative Case Studies
>
> > **W3. Lack of Qualitative Case Studies:**
>
> Thank you for this insightful comment. Based on your suggestion, we have added some qualitative case studies to the appendix of the revised paper (Appendix D). For convenience, we present one representative example here (candidates generated by Qwen-7B).
>
> **question:**
> *“List out the accounts who have the earliest trading date in 1995.”*
>
> **Our method selects:**
> SELECT account_id
> FROM trans
> WHERE date LIKE '1995%'
> ORDER BY date ASC
> LIMIT 1;
>
> **MBMBR selects:**
> SELECT DISTINCT account_id
> FROM trans
> WHERE `date` BETWEEN '1995-01-01' AND '1995-12-31'
> ORDER BY `date` ASC
> LIMIT 1;
>
> The MBMBR-selected query is incorrect because the presence of DISTINCT prevents identification of the earliest transaction when an account has multiple entries in 1995. Across the 32 candids, 12 candidates contain the incorrect DISTINCT pattern while 8 candids are correct. Correct queries have higher probability so our method can successfully select them using the proper integration of probabilities.

---

> ### Author Response · Authors · 2025-11-21
> **Response to Reviewer TztK (Part 3)**
>
> ## Response to Weakness W4: Minor Presentation Error
>
> > **W4. Minor Presentation Error: The illustrative example in Section 3.1 has a distracting error. The probabilities given (, , ) sum to 1.8, not 1. This is a minor but sloppy mistake that should be corrected.**
>
> Thank you for pointing out this error. We changed the values in the examples so that the probabilities sum to one.
>
> ---
>
> ## Response to Q1: Cost–Performance Trade-off
>
> > **Q1. Cost-Performance Trade-off:**
> > Could the authors provide a detailed trade-off analysis comparing latency/cost versus accuracy, given that marginal probability estimation requires multiple LLM calls per candidate?
>
> Thank you for raising this important question. Based on your comment, we have added a section discussing the cost-performance trade-off. We have also provided a detailed explanation of our cost analysis in our response to **W2**.
>
> In summary, our method is absolutely practical. The cost of our method is almost identical to baselines costs for small LLMs (lower bound), and it is p times higher than baselines costs for larger models (upper bound) while this increase in cost can be tackled by parallelization to achieve an identical response time to baseline methods. (p is the number of prompts to estimate the marginal probability and it is equal to 5 in the paper)
>
> ---
>
> ## Response to Q2: Utility Function Motivation
>
> > **Q2. Utility Function Motivation:**
> > What is the theoretical or empirical motivation for the specific piecewise exponential design of the utility function in Eq. 5? Why this particular form over simpler alternatives?
>
> Thank you for your question. As mentioned in our response to **W1**, we have expanded the discussion of the utility function in a newly added subsection in the Method section. This subsection now provides a more detailed explanation of the motivation behind the design choices in Eq. 5.
>
> **Motivation for the Piecewise Exponential Design.** The structure of the utility function reflects the semantics of SQL execution results.
> - (1) Empty outputs and execution errors represent complete failure and should receive uniformly low utility,
> - (2) partially valid outputs (high NULL ratios) represent degraded results, and
> - (3) valid outputs warrant a continuous similarity-based score.
>
> A piecewise function therefore matches the discrete nature of execution correctness. We use exponentials because they integrate cleanly with log-probabilities used during decoding and naturally create smooth yet well-separated utility scales.
>
> ---
>
> ## Response to Q3: Utility Function Motivation and Sensitivity
>
> > **Q3.** The utility function in Eq. 5 appears complex and relies on several hard-coded values. What is the theoretical or empirical motivation for this design? Why are the penalties set to \(e^{-2}\) and \(e^{-1}\)? Could the authors provide a sensitivity analysis for the tunable hyperparameters \(\epsilon\) and \(\lambda\)? This is crucial to alleviate concerns about overfitting.
>
> Thank you for this insightful question. We have already discussed the constants $e^{-2}$, $e^{-1}$, and $\epsilon$ in our response to **W1**. Additionally, we have discussed the sensitivity analysis for $\lambda$ in our response to **W1**. For more detail answer please see our response to **W1**.
>
> In summary, $e^{-2}$ and $e^{-1}$ constants are chosen to provide an order between invalid < partially valid < valid queries. these constants can be any values as long as the order is preserved (minimum score for valid queries is $e^0$). $\epsilon$ is chosen based on an empirical observation in training data. $\lambda$ is our only tunable parameter and we have provided its sensitivity analysis in the paper and in our response to **W1**
>
> ---
>
> ## Response to Q4: Qualitative Analysis
>
> > **Q4. Qualitative Analysis:**
> > Could the authors provide a concrete case study from BIRD or SPIDER showing a real instance where MBMBR selects an incorrect low-probability query, and how the proposed scoring function corrects it?
>
> Thank you for this insightful question. In response, we have added detailed qualitative case studies to the appendix section of our paper. In our response to **W3**, a case study is presented for your reference. A real example from the BIRD benchmarks illustrating how MBMBR may select a wrong but mutually similar low-probability candidate, and how our proposed scoring function in Eq. (4) successfully identifies the correct high-probability query.
>
> We kindly refer the reviewer to our explanation and example provided in the response to **W3**, where we present a qualitative analysis.

---

### Author Response · Authors · 2025-12-02
**Summary Response**

We thank the AC for overseeing the review process and for the opportunity to respond. Below is a concise summary of how we addressed all reviewer concerns and strengthened the paper accordingly.

---

## ✔️ Main Contributions Acknowledged by Reviewers
Across reviewers, several positive aspects of the paper were consistently highlighted:
- The paper identifies **two important flaws** in existing ensemble decoding methods (MBR, MBMBR):
  **(1) prompt-sensitivity bias**, and **(2) neglecting a candidate’s own probability**.
- The proposed method is considered **technically sound**, **well-motivated**, and **clearly formulated**.
- Experiments were found **comprehensive**, covering BIRD, SPIDER, and multiple LLM families.
- Ablations clearly separate the contributions of each component.

We thank the reviewers for recognizing the novelty and practical value of the contributions.

---

## ✔️ Summary of Major Revisions Made in Response to Reviewer Feedback

### **1. Hyperparameter Discussion + Sensitivity Analysis Added**
In response to critiques of Eq. (5), we:
- Added a new subsection thoroughly explaining the role and motivation of all hyperparameters.
- Clarified that constants such as $e^{-2}$ and $e^{-1}$ merely enforce ordering (invalid < partially valid < valid) and **do not require tuning**
- Explained that the threshold $\epsilon = 0.2$ is empirically grounded in training data.
- Added a **new sensitivity analysis for the only tunable parameter $\lambda$** (now in Appendix C), showing:
  - Our method is robust across a wide range of values.
  - It consistently outperforms MBMBR *for all tested $\lambda$*.

This addresses concerns about “brittleness” and possible overfitting.

---

### **2. Cost–Performance Trade-off Section Added**
We added a detailed **latency and cost analysis** in Section 4.2, clarifying:

- The total runtime of our method is *identical* to MBR/MBMBR when using small LLMs (execution time dominates).
- For large LLMs, the additional cost scales linearly with the number of prompts \(p\), *but*:
  - \(p\) is fully user-controllable (2–3 already provides large stability gains).
  - All probability-estimation calls can be **parallelized**, keeping real-world latency unchanged.
- Thus the method is **practical for both small and large models**, with tunable cost.

This directly addresses concerns about the cost-benefit ratio.

---

### **3. Added Qualitative Case Studies (Appendix D)**
To strengthen motivation and support the “reward hacking” claim:
- We added real examples from BIRD showing:
  - MBMBR selecting an incorrect **low-probability but mutually similar** SQL query.
  - Our method correctly selecting the **high-probability correct** query through probability-aware scoring.
- A representative example is also included in our response to reviewers.

This addresses requests for concrete evidence of problematic MBMBR behavior.

---

### **4. Corrected Presentation Issues**
- Fixed probability inconsistency in Section 3.1.
- Improved clarity of Section 3.1 and 3.2 based on reviewer suggestions.

---

### **5. Additional Clarifications for Reviewer N3kd**
- We elaborated in our response that standard MBR does not take advantage of the probability signal of each candidate, which is the core limitation addressed by both MBMBR and our proposed method.
- We highlighted that our experimental results demonstrate **statistically significant performance gains** for most models across both datasets.
- We directed the reviewer to our **ablation study**, which details how each component of our method contributes to the observed improvements.
- We clarified that our method is fundamentally different from prior MBR extensions that they mentioned in their review, and is in fact **complementary** to them. Therefore, our approach can potentially be combined with these methods to yield further performance improvements.
- We thanked the reviewer for noting that our method appears applicable to other tasks. However, we emphasized that the current paper is focused exclusively on the Text-to-SQL task. To make this clear, we updated the conclusion section to state explicitly that extending our method to other tasks is a direction for future work.
- Additionally, we clarified that the scope of this paper is limited to **heuristic-based ensemble methods**, and our contributions aim specifically at improving these methods. We noted in the conclusion that comparing our method against **fine-tuned approaches** is also an avenue for future investigation.

---

## ✔️ Overall
The final version:
- Adds new **qualitative case studies**
- Adds new **hyperparameter analyses**
- Adds a **detailed cost–accuracy trade-off** section
- Clarifies the motivation and theory behind the scoring and utility functions
- Improves correctness, presentation, and experimental transparency

We respectfully submit that the revised paper addresses all reviewers' concerns thoroughly.

---

### Meta-Review · Area_Chair_QgtM · 2026-01-07

**Summary:**

This paper proposes an improved version of Minimum Bayes Risk (MBR) decoding algorithm to address the text-to-SQL problem. The work is well-motivated, and reviewers generally agree that the paper addresses a real limitation of existing MBR and Model-Based MBR (MBMBR) methods, particularly reward hacking and prompt sensitivity. However, reviewers also raise several concerns that limit the strength of the submission.

A primary concern is the practicality and robustness of the proposed method. Multiple reviewers question whether the performance gains (often modest in absolute terms) justify the additional computational cost introduced by marginal probability estimation. Although the rebuttal provides a detailed cost–performance analysis, the concerns about the method’s suitability for real-time or cost-sensitive deployments remain.

Reviewers also express concerns regarding hyperparameter complexity and sensitivity. While the rebuttal adds a dedicated discussion and sensitivity analysis, the utility function itself is designed as heuristic-heavy and insufficiently grounded theoretically. Relatedly, although ablation studies clarify where improvements come from, the theoretical contribution is generally incremental.

Finally, reviewers note limitations in scope and generalization. The evaluation is restricted to text-to-SQL, and while this is a strong and structured testbed, reviewers question whether the method’s benefits extend to other generation tasks where MBR-style ensembles are used. These limitations are acknowledged by the authors, but broader validation is deferred to future work.

Overall, the paper presents a solid and careful empirical refinement of MBR decoding for text-to-SQL. Reviewers agree that it improves over existing heuristic ensembles, but the gains, theoretical depth, and practical trade-offs are not sufficient for acceptance at ICLR.

**Reviewer Concerns:**

I believe that some reviewer concerns are partially addressed by the rebuttal, while others remain outstanding.

Reviewer TztK’s concerns regarding hyperparameter sensitivity, cost–performance trade-offs, and lack of qualitative examples are largely addressed through added sensitivity analyses, runtime modeling, and concrete case studies. However, concerns about overall practicality and cost-effectiveness for real-time deployment may still remain.

Reviewer N3kd’s concerns about small absolute performance gains, limited task generalization, and unclear sources of improvement are partially mitigated by ablations, statistical significance analysis, and qualitative examples. Nonetheless, concerns about broader applicability beyond text-to-SQL and reason of improvement would likely remain.

Reviewer DCDh concerns about limited theoretical depth and modest gains. While additional analysis provided, the theoretical contribution remains relatively lightweight.

**Reviewer Scores:**

Across reviewers, a full discussion would likely result in a small positive score adjustments, primarily due to the improved clarity, added sensitivity analyses, and cost–performance discussion, and qualitative case studies introduced in the rebuttal. However, remaining concerns about computational cost, generalization beyond text-to-SQL, and the incremental nature of the theoretical contribution suggest that some reviewers would likely maintain a cautious stance.

---

### Decision · Program_Chairs · 2026-01-26

Reject